# Correlation of Nutritional Indices on Admission to the Coronary Intensive Care Unit with the Development of Delirium

**DOI:** 10.3390/nu10111712

**Published:** 2018-11-08

**Authors:** Yurina Sugita, Tetsuro Miyazaki, Kazunori Shimada, Megumi Shimizu, Mitsuhiro Kunimoto, Shohei Ouchi, Tatsuro Aikawa, Tomoyasu Kadoguchi, Yuko Kawaguchi, Tomoyuki Shiozawa, Kiyoshi Takasu, Masaru Hiki, Shuhei Takahashi, Katsuhiko Sumiyoshi, Hiroshi Iwata, Hiroyuki Daida

**Affiliations:** 1Department of Cardiovascular Medicine, Juntendo University Graduate School of Medicine, 2-1-1 Hongo Bunkyo-ku, Tokyo 113-8421, Japan; yrsugita@juntendo.ac.jp (Y.S.); shimakaz@juntendo.ac.jp (K.S.); megumi-s@juntendo.ac.jp (M.S.); mkunimo@juntendo.ac.jp (M.K.); uchi@juntendo.ac.jp (S.O.); taikawa@juntendo.ac.jp (T.A.); t-kadoguchi@juntendo.ac.jp (T.K.); yukawagu@juntendo.ac.jp (Y.K.); t-shio@juntendo.ac.jp (T.S.); ktakasu@juntendo.ac.jp (K.T.); ma-hiki@juntendo.ac.jp (M.H.); syutaka@juntendo.ac.jp (S.T.); h-iwata@juntendo.ac.jp (H.I.); daida@juntendo.ac.jp (H.D.); 2Department of Health and Nutrition, Faculty of Human Sciences, Tokiwa University, 1-430-1, Miwa Mito, Ibaraki 310-5385, Japan; kazus@tokiwa.ac.jp

**Keywords:** delirium, malnutrition, acute cardiovascular disease, coronary care unit

## Abstract

Background: Delirium is a common occurrence in patients admitted to the intensive care unit and is related to mortality and morbidity. Malnutrition is a predisposing factor for the development of delirium. Nevertheless, whether the nutritional status on admission anticipates the development of delirium in patients with acute cardiovascular diseases remains unknown. Objective: This study aims to assess the correlation between the nutritional status on admission using the nutritional index and the development of delirium in the coronary intensive care unit. Design: We examined 653 consecutive patients (mean age: 70 ± 14 years) admitted to the coronary intensive care unit of Juntendo University Hospital between January 2015 and December 2016. We evaluated three nutritional indices frequently used to assess the nutritional status, i.e., Geriatric Nutritional Risk Index (GNRI), Prognostic Nutritional Index (PNI), and Controlling Nutritional Status (CONUT). We defined delirium as patients with a delirium score >4 using the Intensive Care Delirium Screening Checklist. Results: Delirium was present in 58 patients. All nutritional indices exhibited a tendency for malnutrition in the delirium group compared with the non-delirium group (GNRI, 86.5 ± 9.38 versus 91.6 ± 9.89; PNI, 36.4 ± 6.95 versus 41.6 ± 7.62; CONUT, 5.88 ± 3.00 versus 3.61 ± 2.56; for all, *p* < 0.001). Furthermore, the maximum delirium score increased progressively from the low- to the high-risk group, as evaluated by each nutritional index (GNRI, PNI, CONUT; for all, *p* < 0.001). A multivariate analysis revealed that the PNI and CONUT were independent risk factors for the occurrence of delirium. Conclusions: A marked correlation exists between the nutritional index on admission, especially PNI and CONUT, and the development of delirium in patients with acute cardiovascular diseases, suggesting that malnutrition assessment upon admission could help identify patients at high risk of developing delirium.

## 1. Introduction

Delirium is a transient neurocognitive disorder that is characterized by inattention, cognitive dysfunction, and behavioral abnormalities, which almost always develop in association with an underlying medical condition. Delirium typically has an acute onset and exhibits a fluctuating course [1]. Delirium is a common occurrence in patients who are admitted to intensive care units. Several studies have assessed the prevalence of delirium in various populations. One study found that one-third of all general medical patients aged ≥70 years have delirium [1]. Delirium is the most common surgical complication among older adults and has an incidence of 15–25% after major elective surgery and up to 50% after high-risk procedures such as hip fracture repair and cardiac surgery [2]. In a point-prevalence study conducted across 108 acute care units and 12 rehabilitation wards in Italian hospitals on the same day, 429 (22.9%) out of 1867 older patients were found to have delirium [3]. Delirium was shown to be present in 10–20% of older adults in the emergency unit [4]. Among 611 patients with heart failure, 139 patients (23%) experienced delirium during their hospital stay [5].

Delirium was also shown to be associated with poor clinical outcomes, including higher emotional burden for patients and their caregivers, decreased cognitive and functional performance, a high rate of discharge to places other than the home, and high short- and long-term mortality [3]. However, the pathophysiological mechanisms of delirium have not been fully elucidated [1]. The development of delirium is linked to several risk factors, including old age, poor activity-of-daily-living scores, dementia, inflammatory state, use of antipsychotic drugs, feeding tubes, and peripheral venous and urinary catheters, physical restraints, and malnutrition [3,6]. In particular, nutritional deprivation was shown to be associated with early postoperative delirium in patients undergoing surgical procedures [7,8].

Malnutrition is prevalent in older adults, particularly those with chronic disease, and may lead to sarcopenia and frailty [9]. The stressors of acute-on-chronic diseases, such as exacerbations of chronic heart failure, increase the susceptibility to accelerated weight loss via mechanisms such as muscle protein catabolism, inactivity, and counterregulatory hormone surges [10]. This combination of multiple physiologic insults can breach the threshold of frailty and result in physical and functional decline and loss of independence [11]. Recent studies have shown that delirium is one of the most frequent presentations of frailty and indicates a vulnerable brain [6].

The assessment of the nutritional status is a crucial aspect of the care of patients with cardiovascular diseases [12]. Moreover, delirium is related to long-term mortality and morbidity, particularly in patients with acute heart diseases [5,13]. Nevertheless, the clinical significance of nutritional indices and their association with the development of delirium in patients with acute cardiovascular disease remains unclear. Thus, the current study investigated whether the nutritional status at admission predicts the development of delirium in patients with acute cardiovascular diseases.

## 2. Materials and Methods

### 2.1. Study Design

This study was part of an ongoing cohort study of biomarkers in patients admitted to a coronary intensive care unit (UMIN-CTR; UMIN000007555); however, we collected data prospectively with a systematic approach. We enrolled 653 consecutive patients (mean age: 70 ± 14 years) who were admitted to the coronary care unit (CCU) of Juntendo University Hospital (Tokyo, Japan) between January 2015 and December 2016. Patients who were receiving respiratory support or who had severe disturbance of consciousness were excluded because of difficulties in assessing their delirium score. The enrolled patients were categorized into delirium and non-delirium groups. By using the Intensive Care Delirium Screening Checklist, delirium was defined as a delirium score >4 [14]. The delirium scores were assessed by independent nurses upon admission and three times a day until the discharge of the patients from the CCU. The patients in the delirium group were defined as the patients who had shown a delirium score >4 at least once during their stay at the CCU.

We defined acute decompensated heart failure (ADHF) on the basis of the diagnostic criteria of the Framingham Heart Study [15]. Furthermore, acute coronary syndrome was defined as the presence of unstable angina pectoris, non-ST-elevation, or ST-elevation myocardial infarction [16]. Diabetes mellitus was defined on the basis of a prior diagnosis of diabetes mellitus (from medical records), hemoglobin A1c (HbA1c; national glycohemoglobin standardization program calculation) levels >6.5%, or current treatment with antidiabetic agents or insulin. Dyslipidemia was defined on the basis of a prior diagnosis of dyslipidemia (from medical records), presence of an abnormal lipid profile (i.e., triglyceride [TG] levels > 150 mg/dL; low-density lipoprotein cholesterol [LDL-C] > 140 mg/dL; or high-density lipoprotein cholesterol [HDL-C] < 40 mg/dL), or current treatment with antidyslipidemic agents. Hypertension was defined on the basis of a prior diagnosis of hypertension (systolic blood pressure > 140 mmHg or diastolic blood pressure > 90 mmHg) from medical records or current treatment with antihypertensive agents. The study protocol was approved by the Ethics Committee of Juntendo University Hospital. Written informed consent was obtained from all participants prior to their enrollment.

### 2.2. Blood Sampling

Fasting blood samples were drawn from all patients within 24 h of admission. Plasma total cholesterol (TC), TG, and HDL-C were assessed using standard enzymatic methods, and LDL-C was evaluated using the Friedewald formula. Plasma glucose concentration, HbA1c, C-reactive protein (CRP), and creatinine levels were assessed by standard methods.

### 2.3. Assessment of Nutritional Status

We assessed the nutritional status at admission using three recently documented scoring systems to estimate the values of nutrition-related prognostic risk factors. The Geriatric Nutritional Risk Index (GNRI) is a simple and accurate tool for the estimation of the risk of morbidity and mortality in hospitalized elderly patients [17]. GNRI was shown to be significantly associated with Mini Nutritional Assessment Long and Short forms and with the mortality risk of patients admitted to geriatric acute wards [18]. GNRI is calculated using the following formula: 14.89 × serum albumin concentration (g/dL) + 41.7 × (body weight/ideal body weight). The patients were categorized into the following grades of nutrition-related risk: major risk (GNRI < 82), moderate risk (GNRI = 82–92), low risk (GNRI = 92–98), and no-risk (GNRI > 98). The Prognostic Nutrition Index (PNI), which is mainly used for Asian populations, is a simple and objective indicator of postoperative outcomes in cancer patients who are undergoing surgery [19,20,21,22,23]. A recent study demonstrated the relevance of PNI for the assessment of patients with acute heart failure [24]. PNI is calculated using the following formula: 10 serum albumin concentration (g/dL) + 0.005 total lymphocyte count (number/mm^2^) in peripheral blood. Patients with a PNI score >38 are considered normal, those with a score of 35–38 are at moderate risk of malnutrition, and those with a score <35 are considered at severe risk [25]. The Controlling Nutritional Status (CONUT) method is an automated malnutrition screening and alert tool for patients hospitalized with acute diseases [26]. Moreover, serum albumin, TC, and total lymphocyte count were used as indicators of storage and availability of protein stores [27], energy reserves [28], and impaired immune defense, respectively, and are closely linked to malnutrition [29]. The scoring criteria (maximum of 12 points) were as follows: 0–4 points, low risk; 5–8 points, moderate risk; and 9–12 points, high risk.

### 2.4. Statistical Analysis

From our preliminary data, the mean occurrence rate of delirium in patients admitted to the CCU was approximately 9% (unpublished data). In this study, we hypothesized that malnourished patients have twice the occurrence rate of delirium compared with patients with good nutritional status (18% versus 9%, respectively). According to statistical power analysis, a minimum sample size of 450 patients was required to detect a substantial relative risk reduction with 80% statistical power and a two-sided α of 5%. Continuous variables are expressed as mean ± standard deviation, and categorical variables are presented as percentages. Furthermore, we compared continuous variables using the Mann–Whitney U-test and analyzed categorical variables using the chi-squared test. Univariate logistic regression analysis was performed to assess the effect of parameters at admission on the occurrence of delirium. Subsequently, multivariate logistic regression analysis was performed to determine the effects of each nutritional index on the occurrence of delirium. Model 1 was adjusted for age, gender, and body mass index, whereas models 2 and 3 were adjusted for the variables of model 1, along with the risk factors obtained from the results of the univariate logistic regression analysis (including ADHF at admission, history of dementia, CRP level, and albumin level). When the association between each nutritional index and delirium was analyzed, we eliminated patients for whom the data required for the calculation of GNRI, PNI, and CONUT were missing (*n* = 20, 17, and 55, respectively) from each analysis. All statistical analyses were performed using JMP (version 12.0 for Windows; SAS Institute, Cary, NC, USA); *p* < 0.05 was considered statistically significant. 

## 3. Results

In this study, we enrolled 653 consecutive patients (mean age: 70 ± 14 years) who were admitted to the CCU. The mean follow-up duration was 13.0 ± 20.7 days. Among the 653 patients, 439 (67%) were male, and 58 patients (8.9%) developed delirium. The age of the patients in the delirium group was significantly higher, whereas the body mass index and the percentage of males was significantly lower than in the non-delirium group. Furthermore, the delirium group had a higher number of patients with a history of dementia. More than half of all patients in the delirium group were admitted with ADHF. Furthermore, albumin, TC, HDL-C, and LDL-C levels were significantly lower, whereas creatinine, CRP, and N-terminal pro-brain natriuretic peptide (NT-proBNP) levels were significantly higher in the delirium group. Notably, a greater proportion of patients in the delirium group received antipsychotic drugs (Table 1). Each nutritional index was moderately and strongly associated with several parameters, including older age, low body mass index, low albumin levels, low TC and LDL-C levels, and inflammatory state (Appendix A).

Each of the nutritional indexes exhibited a tendency for malnutrition in the delirium group compared with the non-delirium group (GNRI: 86.5 ± 9.38 versus 91.6 ± 9.89; PNI: 36.4 ± 6.95 versus 41.6 ± 7.62; CONUT: 5.88 ± 3.00 versus 3.61 ± 2.56, respectively; *p* < 0.001 for all; Figure 1). Furthermore, the maximum delirium score increased progressively from the no-risk category to the high-risk category, as assessed by each nutritional index (Figure 2). Univariate logistic regression analysis showed that higher age, female sex, low body mass index, low albumin, TC, HDL-C, and LDL-C levels, high CRP, NT-proBNP, creatinine levels, ADHF at admission, and history of dementia and antipsychotic use were found to be significant risk factors for the occurrence of delirium (Table 2). Multivariate logistic regression analysis showed that PNI and CONUT were independent risk factors for the development of delirium (Table 3).

## 4. Discussion

In this study, malnutrition at admission showed a significant correlation with the development of delirium in patients with acute cardiovascular diseases who were admitted to the CCU. Our findings suggest that the early assessment of the nutritional status may help identify patients who are at high risk of delirium. This study demonstrates that malnutrition is an independent predictor of the development of delirium in patients with acute cardiovascular diseases.

Previously, Ringaitiene et al. [7] demonstrated the correlation between malnutrition and postoperative delirium in patients after on-pump coronary artery bypass grafting. Likewise, another study showed a correlation between preoperative malnutrition and postoperative delirium after hip fracture surgery in older patients [8]. In this study, we observed a correlation of malnutrition at admission with the development of delirium in patients with acute cardiovascular diseases, which was relatively unstable compared to the preoperative status.

Given that nutrition is vital for all functions of the body and that the brain is an organ with high metabolic activity and nutritional demands, it is entirely plausible that nutrients affect the development of delirium [7]. The deficiency of micronutrients and vitamins such as niacin and thiamine was shown to affect the onset of delirium because of impaired neurotransmission [30]. However, whether nutrients play a role in delirium is poorly understood. To date, only a few studies have supported the theory that delirium is associated with malnutrition [31]. Although this study also found a marked correlation between malnutrition and the development of delirium in patients with acute cardiovascular diseases, the underlying mechanisms remain uncertain. Thus, further interventional studies are warranted to identify nutrients that may alleviate delirium in patients with acute cardiovascular diseases.

In this study, over half of the patients assigned to the delirium group were admitted with a diagnosis of ADHF. Patients with heart failure often experience a severe loss of muscle strength (i.e., “cachexia”) and alterations in body composition, which are liable to worsen malnutrition [32]. Furthermore, heart failure and cachexia often coexist, and the body composition alterations that occur in heart failure reflect a complicated phenomenon that involves the interplay of several factors, including dietary intake, gastrointestinal absorption, and immunological catabolic and anabolic states [33,34]; this combination of multiple physiological alterations can cause the substantial physical and functional decline of the nutritional status and could be associated with the development of delirium, particularly in patients with heart failure.

Inflammation is an established risk factor for delirium [35]. In this study, multivariate logistic regression analysis revealed that the PNI and CONUT (which include lymphocyte count in their definition, unlike GNRI) predicted the development of delirium even after adjustment for CRP levels; this finding suggests a correlation between the inflammatory state at admission and the development of delirium in the CCU. Furthermore, PNI and CONUT at admission appear to be suitable predictors of the occurrence of delirium because both represent malnutrition and inflammation simultaneously.

Taken together, it is important to improve the nutritional status of patients with acute cardiovascular diseases from an early stage after their admission. The provision of a balanced diet and nutrient supplements may be an effective approach. In particular, antioxidant supplementation (i.e., vitamin E, vitamin C, carotenoids, flavonoids, and omega-3 fatty acids) may improve the cognitive function and decrease the risk of delirium by suppressing the release of inflammatory mediators, which can cause oxidative stress and neuronal damage [31]. Furthermore, essential amino acid supplementation in conjunction with exercise or rehabilitation regimens may confer a direct anabolic effect and have a potentially beneficial role in patients who are at high risk of developing delirium [34].

In this study, delirium developed in approximately 9% of all participants. This percentage is comparatively lower than in previous studies. Previous studies indicated that the development of delirium was linked to old age, poor activity of daily living, dementia, inflammatory state, antipsychotic use, feeding tubes, peripheral venous and urinary catheters, physical restraint, and malnutrition [3,6]. In the current study, old age, history of dementia, use of antipsychotics, inflammatory state, and malnutrition were some of the common risk factors. On the contrary, female sex and ADHF were specific risk factors in the population of the current study. These differences may be attributable to the characteristics of study subjects. The participants in our study included relatively younger patients and had a wide variety of disease types and severity.

This study has several limitations. First, the study was conducted in a single institution with a relatively small sample size. Therefore, studies with a larger sample size are required to investigate the correlation between each nutritional index and the development of delirium in patients with cardiovascular diseases. Second, we did not assess cognitive function and delirium by using other methods (i.e., formal cognitive screening, confusion assessment method, and Delirium Rating Scale-Revised-98), even though we collected data pertaining to the history of dementia and use of antipsychotics from medical records. Therefore, further studies are required to confirm the association between malnutrition and the development of delirium by using various scales. Third, serum albumin levels in patients with cardiovascular diseases may have been affected by the body fluid volume because the blood samples were drawn during the acute phase. Therefore, the three nutritional indices used in this study might be less accurate than those calculated in the chronic phase. Finally, we could not collect data on the dietary status and physical activity of the patients. We could not ascertain the correlation of each nutritional index with the dietary status and physical activity during admission to the CCU.

## 5. Conclusions

In this study, malnutrition at admission was markedly related to the development of delirium in patients with acute cardiovascular diseases admitted to the CCU. Our findings suggest that the assessment of the nutritional status upon admission may help identify patients who are at high risk of developing delirium and result in better care. Nevertheless, further research is warranted to elucidate whether nutritional treatment can prevent the development of delirium and result in better prognoses for patients with cardiovascular diseases.

## Figures and Tables

**Figure 1 nutrients-10-01712-f001:**
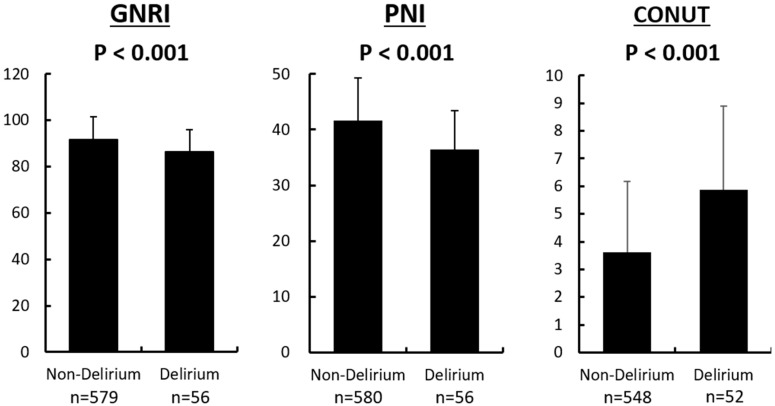
A comparison of the nutritional indices between the delirium and non-delirium groups. Each nutritional index exhibited a tendency toward malnutrition in the delirium group compared with the non-delirium group. Data are presented as mean ± standard deviation. GNRI, Geriatric Nutritional Risk Index; PNI, Prognostic Nutritional Index; CONUT, Controlling Nutritional Status.

**Figure 2 nutrients-10-01712-f002:**
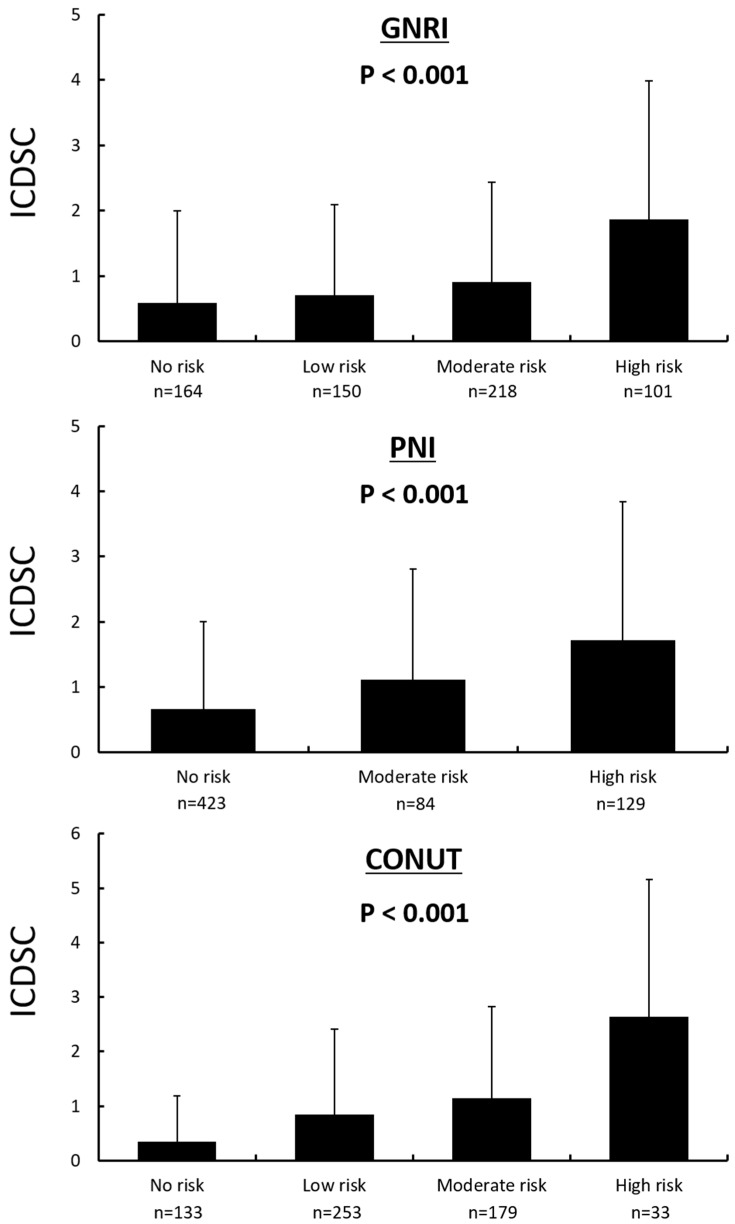
A comparison of the delirium scores of patients with different nutritional status on admission. The delirium score (ICDSC, Intensive Care Delirium Screening Checklist) increased progressively from the no-risk to the high-risk category, as assessed by each nutritional index. Data are presented as mean ± standard deviation.

**Table 1 nutrients-10-01712-t001:** Characteristics of the study subjects.

	Delirium Group (*n* = 58)	Non-Delirium Group (*n* = 595)	*p*
Age, years	80.4 ± 11.1	69.1 ± 13.8	<0.001
Male, *n* (%)	31 (53)	408 (69)	0.03
Body mass index, kg/m^2^	22.4 ± 3.8	23.7 ± 4.5	0.02
Left ventricular ejection fraction, %	55 ± 14	55 ± 16	NS
Diabetes mellitus, *n* (%)	24 (41)	172 (29)	NS
Dyslipidemia, *n* (%)	20 (34)	272 (46)	NS
Hypertension, *n* (%)	35 (60)	305 (51)	NS
Atrial fibrillation, *n* (%)	15 (26)	93 (16)	NS
Dementia, *n* (%)	17 (29)	12 (2)	<0.001
Cerebral infarction, *n* (%)	7 (12)	46 (7.7)	NS
Malignancy, *n* (%)	11 (19)	66 (11)	NS
**Diagnosis on admission**			0.002
Acute decompensated heart failure, *n* (%)	32 (55)	209 (35)	
Acute coronary syndrome, *n* (%)	7 (12)	200 (34)	
Aortic disease, *n* (%)	5 (9)	17 (3)	
PTE/DVT, *n* (%)	2 (3)	20 (3)	
VT/VF, *n* (%)	1 (2)	18 (3)	
Others, *n* (%)	11 (19)	131 (22)	
**Laboratory data**			
Albumin, g/dL	3.2 ± 0.6	3.5 ± 0.6	<0.001
Total cholesterol, mg/dL	149 ± 51	164 ± 41	0.002
Triglycerides, mg/dL	87 ± 56	95 ± 56	NS
HDL-C, mg/dL	40 ± 13	44 ± 14	0.02
LDL-C, mg/dL	89 ± 34	101 ± 33	0.007
Creatinine, mg/dL	1.9 ± 2.2	1.5 ± 1.8	0.003
HbA1c, %	6.4 ± 2.0	6.1 ± 1.0	NS
CRP, mg/dL	4.1 ± 5.9	2.0 ± 4.0	<0.001
NT-pro BNP, pg/mL	14718 ± 22640	6209 ± 15418	<0.001
**Medication**			
Antiplatelets, *n* (%)	17 (29)	222 (37)	NS
Anticoagulants, *n* (%)	14 (24)	101 (17)	NS
ACE-I/ARBs, *n* (%)	20 (34)	195 (33)	NS
β-blockers, *n* (%)	20 (34)	184 (31)	NS
Calcium channel blockers, *n* (%)	15 (26)	194 (33)	NS
Statin, *n* (%)	16 (28)	199 (33)	NS
Oral hypoglycemic agents, *n* (%)	11 (20)	97 (18)	NS
Insulin, *n* (%)	4 (7)	31 (5)	NS
Antipsychotics, *n* (%)	2 (3.5)	2 (0.3)	0.04
Anti-depressants, *n* (%)	0 (0)	3 (0.5)	NS
Anxiolytic drugs, *n* (%)	0 (0)	9 (1.5)	NS
Benzodiazepines, *n* (%)	1 (1.8)	20 (3.4)	NS
Nonbenzodiazepines, *n* (%)	2 (3.5)	6 (1.0)	NS

Data are presented as means ± SD or number (percentage). PTE, pulmonary thromboembolism; DVT, deep vein thrombosis; VT, ventricular tachycardia; VF, ventricular fibrillation; HDL-C, high-density lipoprotein cholesterol; LDL-C, low-density lipoprotein cholesterol; HbA1c, hemoglobin A1c, national glycohemoglobin standardization program calculation; CRP, C-reactive protein; NT-pro BNP, N-terminal pro brain natriuretic peptide; ACE-I, angiotensin-converting-enzyme inhibitor; ARBs, angiotensin II receptor blockers.

**Table 2 nutrients-10-01712-t002:** Univariate logistic regression analyses for the onset of delirium.

	OR	95% CI	*p*
Age, 1 year increase	1.09	1.06–1.12	<0.001
Female	1.90	1.10–3.27	0.02
Body mass index, 1 increase	0.94	0.88–0.99	0.04
Albumin, 1mg/dL increase	0.41	0.25–0.66	<0.001
Total cholesterol, 1 mg/dL increase	0.99	0.98–0.99	0.009
HDL-C, 1 mg/dL increase	0.98	0.96–0.99	0.03
LDL-C, 1 mg/dL increase	0.99	0.98–0.99	0.007
CRP, 1mg/dL increase	1.08	1.03–1.13	0.002
NT-proBNP, 1pg/mL increase	1.00	1.00–1.00	0.005
Creatinine, 1 mg/dL increase	0.90	0.81–1.03	0.051
ADHF on admission	2.27	1.32–3.94	0.003
History of dementia	20.1	9.08–46.0	<0.001
History of diabetes mellitus	1.73	1.00–3.00	0.054
Antipsychotics use	10.8	1.27–91.1	0.03

OR: odds ratio, 95% CI: 95% confidence interval; HDL-C, high-density lipoprotein cholesterol; LDL-C, low-density lipoprotein cholesterol; CRP, C-reactive protein; NT-pro BNP, N-terminal pro brain natriuretic peptide; ADHF, acute decompensated heart failure.

**Table 3 nutrients-10-01712-t003:** Multivariate logistic regression analyses for the occurrence of delirium.

	Crude			Model 1			Model 2			Model 3		
	OR	95% CI	*p*	OR	95% CI	*p*	OR	95% CI	*p*	OR	95% CI	*p*
GNRI, 1 decrease	**1.05**	**1.02–1.07**	**<0.001**	1.03	0.99–1.07	0.06	1.03	0.99–1.07	0.13	0.96	0.84–1.09	0.63
GNRI as a categorical variable												
No risk	1.00	(reference)		1.00	(reference)		1.00	(reference)		1.00	(reference)	
Low risk	1.60	0.60–4.52	0.35	1.01	0.36–2.94	0.98	1.02	0.35–3.09	0.97	1.34	0.39–4.89	0.65
Moderate risk	2.01	0.85–5.23	0.11	1.10	0.44–3.01	0.84	1.09	0.41–3.15	0.86	1.62	0.37–7.62	0.52
High risk	**5.81**	**2.48–15.3**	**<0.001**	**2.86**	**1.10–8.13**	**0.03**	2.46	0.87–7.48	0.09	3.93	0.47–33.0	0.20
PNI, 1 decrease	**1.10**	**1.06–1.14**	**<0.001**	**1.08**	**1.03–1.13**	**<0.001**	**1.07**	**1.02–1.13**	**0.003**	**1.17**	**1.05–1.33**	**0.004**
PNI as a categorical variable												
No risk	1.00	(reference)		1.00	(reference)		1.00	(reference)		1.00	(reference)	
Moderate risk	1.83	0.74–4.09	0.18	1.08	0.42–2.53	0.86	1.03	0.38–2.51	0.95	1.11	0.37–3.06	0.84
High risk	**4.18**	**2.28–7.71**	**<0.001**	**2.90**	**1.52–5.52**	**0.001**	**2.53**	**1.26–5.04**	**0.009**	2.62	0.89–7.68	0.08
CONUT, 1 increase	**1.33**	**1.20–1.48**	**<0.001**	**1.31**	**1.17–1.48**	**<0.001**	**1.29**	**1.14–1.46**	**<0.001**	**1.44**	**1.17–1.79**	**0.02**
CONUT as a categorical variable												
No risk	1.00	(reference)		1.00	(reference)		1.00	(reference)		1.00	(reference)	
Low risk	**5.32**	**1.51–33.7**	**0.006**	3.52	0.97–22.6	0.06	**5.81**	**1.37–42.4**	**0.01**	**7.03**	**1.52–55.0**	**0.009**
Moderate risk	**8.65**	**2.48–54.7**	**<0.001**	**4.51**	**1.23–29.1**	**0.02**	**6.03**	**1.39–43.9**	**0.01**	**8.66**	**1.58–74.6**	**0.01**
High risk	**27.3**	**6.68–185.0**	**<0.001**	**22.1**	**5.03–155.8**	**<0.001**	**33.2**	**6.39–270.9**	**<0.001**	**46.1**	**4.90–601.4**	**<0.001**

Model 1: adjusted for age, gender, and body mass index; Model 2: adjusted for age, gender, body mass index, acute decompensated heart failure on admission, and history of dementia; Model 3: adjusted for age, gender, body mass index, acute decompensated heart failure on admission, history of dementia, C-reactive protein, and albumin levels; OR, odds ratio; 95% CI, 95% confidence interval. The bold means statistically significant values.

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
