# Peer review of "Correlation of Nutritional Indices on Admission to the Coronary Intensive Care Unit with the Development of Delirium"

_nutrients, 2018, doi:10.3390/nu10111712_

Reviewer 1 Report

Please see my comments below:

1)      Please revise the introduction. The introduction is too brief and does not provide in-depth information about the background of the study.

2)      Introduction: What is the prevalence of delirium in the population?

3)      Introduction: What are the risk factors of delirium?

4)      Did the authors perform the 3 phases of assessment to screen for delirium? (formal cognitive screening, confusion assessment method and Delirium Rating Scale-Revised-98 and DSM-IV assessment)

5)      When did the authors assess delirium in patients after they were admitted to hospital? Were patients assessed within 6 hours after they were admitted to hospital? These details were not provided.

6)      Were the Geriatric 92 Nutritional Risk Index (GNRI) validated in the population? If yes, please report the validity parameters.

7)      Were the Prognostic Nutrition Index 97 ethnic- and gender-specific? Please provide the details for this.

8)      Results: please report the risk factors for delirium and malnutrition.

9)      Discussion: Please compare the risk factors and prevalence of delirium in the study with other published studies. This discussion is missing.

10)  Please improve the English throughout the manuscript.

Reviewer 2 Report

In this manuscript, the authors investigated the correlation between the nutritional status on admission using the nutritional index and the development of delirium in the coronary intensive care unit. The result proved that malnutrition on admission is markedly related to the development of delirium in patients with acute cardiovascular diseases, suggesting that assessing the nutritional status in early stages could help determine patients at high risk of developing delirium and result in better care.

 This research is very meaningful and well designed.
After obtaining the positive results, I suggest the authors add a paragraph to provide some possible nutritional advice to patients with acute cardiovascular diseases.

Author Response

Round  2

Reviewer 1 Report

The authors have revised accordingly and their response met my expectations.

Please see my minor comments as below:

1) Line 35: please indicate which indices.

2) Line 82: revise to " data collected prospectively in a systematic approach"

3) line 135: please provide the reference.

4) line 148: Can the authors perform intention-to-treat for the missing data?

5) line 212: I suggest delete the phrase "this is the first study..."
